# Short-Term Effects of Heat Stress on Cow Behavior, Registered by Innovative Technologies and Blood Gas Parameters

**DOI:** 10.3390/ani14162390

**Published:** 2024-08-18

**Authors:** Ramūnas Antanaitis, Karina Džermeikaitė, Justina Krištolaitytė, Renalda Juodžentytė, Rolandas Stankevičius, Giedrius Palubinskas, Arūnas Rutkauskas

**Affiliations:** 1Large Animal Clinic, Veterinary Academy, Lithuanian University of Health Sciences, Tilžės Str. 18, LT-47181 Kaunas, Lithuania; karina.dzermeikaite@lsmu.lt (K.D.); justina.kristolaityte@lsmu.lt (J.K.); arunas.rutkauskas@lsmu.lt (A.R.); 2Department of Animal Breeding, Veterinary Academy, Lithuanian University of Health Sciences, Tilžės Str. 18, LT-47181 Kaunas, Lithuania; renalda.juodzentyte@lsmu.lt (R.J.); giedrius.palubinskas@lsmu.lt (G.P.); 3Department of Animal Nutrition, Lithuanian University of Health Sciences, Tilzes Str. 18, LT-47181 Kaunas, Lithuania; rolandas.stankevicius@lsmu.lt

**Keywords:** thermal stress, dairy cattle behavior, temperature–humidity index, blood gas analysis, smart farming

## Abstract

**Simple Summary:**

This study examines ways in which heat stress affects the behavior and health of dairy cows using advanced monitoring technologies. Higher levels of heat stress significantly reduce the time cows spend ruminating and increase their body temperature. Furthermore, key blood gas parameters are altered, indicating changes in the cows’ metabolic state. Cows within the highest THI classification group (73–78) showed a reduction in partial carbon dioxide pressure (pCO2), a 32% increase in partial oxygen pressure (pO2), a decrease in sodium by 1.36%, and a decrease in potassium by 6%, while chloride levels increased by 3%. These findings highlight the importance of using innovative technologies to monitor and manage heat stress in dairy cows to ensure their health and productivity. They may also serve as an early indicator, allowing time to implement management changes to reduce the impact of heat stress.

**Abstract:**

Heat stress (HS) is one of the key factors affecting an animal’s immune system and productivity, as a result of a physiological reaction combined with environmental factors. This study examined the short-term effects of heat stress on cow behavior, as recorded by innovative technologies, and its impact on blood gas parameters, using 56 of the 1070 cows clinically evaluated during the second and subsequent lactations within the first 30 days postpartum. Throughout the experiment (from 4 June 2024 until 1 July 2024), cow behavior parameters (rumination time min/d. (RT), body temperature (°C), reticulorumen pH, water consumption (L/day), cow activity (h/day)) were monitored using specialized SmaXtec boluses and employing a blood gas analyzer (Siemens Healthineers, 1200 Courtneypark Dr E Mississauga, L5T 1P2, Canada). During the study period, the temperature–humidity index (THI), based on ambient temperature and humidity, was recorded and used to calculate THI and to categorize the data into four THI classes as follows: 1—THI 60–63 (4 June 2024–12 June 2024); 2—THI 65–69 (13 June 2024–18 June 2024); 3—THI 73–75 (19 June 2024–25 June 2024); and 4—THI 73–78 (26 June 2024–1 July 2024). The results showed that heat stress significantly reduced rumination time by up to 70% in cows within the highest THI class (73 to 78) and increased body temperature by 2%. It also caused a 12.6% decrease in partial carbon dioxide pressure (pCO2) and a 32% increase in partial oxygen pressure (pO2), also decreasing plasma sodium by 1.36% and potassium by 6%, while increasing chloride by 3%. The findings underscore the critical need for continuous monitoring, early detection, and proactive management to mitigate the adverse impacts of heat stress on dairy cow health and productivity. Recommendations include the use of advanced monitoring technologies and specific blood gas parameter tracking to detect the early signs of heat stress and implement more timely interventions.

## 1. Introduction

Heat stress (HS) is a critical factor that impacts an animal’s immune system and the productivity of dairy cows [1]. It occurs when environmental conditions and physiological reactions cause an increase in body temperature, leading to HS when the internal heat generation exceeds a cow’s ability to dissipate heat [2]. The average temperature of the globe has increased by 1.4 °C to 5.8 °C in the twenty-first century, or 0.2 °C per ten years, according to the Intergovernmental Panel on Climate Change [3]. The thermoneutral zone of dairy cattle is the temperature range between 16 and 25 °C, where a healthy adult animal can maintain a normal body temperature without expending additional energy beyond its typical basal metabolic rate [4]. A dairy cow experiences increased heat gain relative to heat loss from the body when it exceeds its thermoneutral zone, which is defined as a surface temperature of 22 to 25 °C in moderate weather and 26 to 37 °C in hot weather. This causes the animal’s core body temperature to rise above the typical range, which leads to hypothermia [5].

Temperature–humidity is a metric that has been widely used to gauge heat stress in dairy cattle [6]. Elevated temperatures and humidity can cause pain and exacerbate stress in cows; this can have a significant effect on the physiology of dairy cows [7]. The temperature–humidity index (THI) is therefore a possible indicator of HS in dairy cattle, and THI readings above 72 may be stressful and have a detrimental impact on the well-being and productivity of dairy cows [1]. An animal’s health and welfare are directly impacted by HS because it lowers a dairy cow’s immune function [1]. Dairy cows under HS are less able to meet energy needs for both milk production and general maintenance of health since energy intake is reduced [8]. This condition lowers milk production and quality, leading to reduced fat and protein content in milk, and increases the animals’ susceptibility to diseases due to lower feed and energy intake [9]. Therefore, farms need to use effective strategies for minimizing HS in dairy cattle, which include dietary interventions and physical modifications to the cow’s environment, such as shade, shelter, and cooling systems, may help to reduce some of the negative effects of HS and may improve the health and productivity of dairy cattle during hot conditions [1].

Precision livestock farming (PLF) is the practice of managing the smallest production units with real-time monitoring technologies while utilizing sensor technology to focus on individual animals, which offers significant opportunities to generate value for many parties involved, primarily serving as an effective instrument for farmers. It simultaneously reduces environmental impacts and enhances animal care, efficiency, and health [10]. When employing automatic milking systems (AMSs) to detect mastitis in herds, it is imperative to use several data sources and alternate methodologies [11]. To identify cows with health difficulties, sensor data can be used either alone or in conjunction with conventional health monitoring techniques [12]. Monitoring behavioral and health markers allows for the early detection of small changes before they become clinical symptoms. Since pre-diagnostic data may more accurately predict risk or diagnose disease than clinical observation alone, it is frequently more important for early identification and care [13]. However, more data are needed to develop protocols for identifying and preventing diseases utilizing information from automated health monitoring systems [14].

In recent times, PLF technology has been able to monitor temperature, milk yield, activity levels, laying, and rumination times [15]. The study’s findings regarding behavioral and physiological differences provide insight into the ways that breed, parity, milk yield (MY), and THI affect cows [15]. Using reticulorumen temperature (bolus sensor), an algorithm was developed to eliminate drinking points from reticular temperature and connect the reticulorumen fermentation temperature with vaginal temperature [16]. By employing the ruminal bolus, this technique enables the accurate online continuous evaluation of a cow’s body temperature and quantifies the link between vaginal and reticular temperature [16]. These boluses are capable of measuring pH and temperature. Every ten minutes, wireless boluses can send data that can be stored on a computer or in the cloud. Measurements can be made for up to a year, based on how long the different bolus versions’ batteries last [16]. Regarding physiological factors, there were stronger positive connections seen between respiratory rate and rectal temperature and both barn temperature and THI [17].

These tissues’ metabolic processes may change under HS circumstances, which could result in variations in oxygen consumption and, as a result, variations in the concentrations of oxygen in the venous blood supply. Moreover, HS may affect vascular and blood flow, which may further explain variations in oxygen content [18]. The equilibrium of electrolyte levels in blood might be impacted by increased sweating and respiratory alkalosis during hypotension [19]. Consequently, variations in Ca2+, Na+, and K+ under varying HS levels may be a sign of modifications in the electrolyte balance and overall health of dairy cows [20]. Milk electrolyte concentrations may provide insight into transient physiological reactions associated with hypotension [20]. These findings suggest that HS significantly impacts the overall health of dairy cows [18].

Since the introduction of devices that automatically record behavior, there has been a significant increase in studies examining the laying and eating habits of dairy cows in relation to housing and management, nutrition, and health. Behavior-recording technologies have been commercially available in recent years, providing new opportunities for precise cow husbandry [21].

This study examined the short-term effects of heat stress on cow behavior, as recorded by innovative technologies, and its impact on blood gas parameters.

## 2. Materials and Methods

### 2.1. Study Animal Housing Conditions

By acquiring a license with approval number PK012858, we were able to conduct this study in accordance with the Lithuanian Law on Animal Welfare and Protection. Lithuania was the location of the research (coordinates: 55.819156, 23.773541). Lithuanian Holstein dairy cows were offered a total mixed ration (TMR) that was formulated to meet their physiological needs according to the nutrient requirements of dairy cattle (NRC) [22] all year round and were housed in free-stall barns with DeLaval ventilation systems (DeLaval Inc., a company based in Tumba, Sweden). Feeding took place every day at 6:00 and 18:00 h, offering a TMR appropriate for multiparous, high-yielding cows according to the NRC [22].

A 620 kg Holstein cow yielding 37 kg of milk per day was fed a diet consisting of 25% corn silage, 5% alfalfa grass hay, 20% grass silage, 15% sugar beet pulp silage, 30% grain concentrate mash, and 5% mineral mix (Table 1). The ration’s chemical composition was as follows: 48.8% dry matter (DM), 28.2% neutral detergent fiber, 19.8% acid detergent fiber, 38.7% non-fiber carbs, 15.8% crude protein, and 1.6 Mcal/kg net lactation energy (Table 2). Milking was performed twice a day using a parlor system, at 5:00 and 17:00. The milking process was carried out using a DeLaval milking parlor manufactured by DeLaval Inc., a company based in Tumba, Sweden.

### 2.2. Selection of Cows for Study

A total of 56 of the 1070 clinically evaluated Lithuanian Holstein cows—that is, those in their second or subsequent lactation and within the first 30 days following calving—were chosen for the study. These cows were selected because they are in a critical period for lactation performance and are more likely to experience heat stress, which could impact their milk production and overall health. Additionally, cows in their second or subsequent lactations provide a consistent basis for evaluating the effects of heat stress due to their established lactation history. These cows weighed an average of 620 kg ± 45 kg, and they produced 12,700 kg of energy-corrected milk per lactation (with 4.2% fat and 3.6% protein). The same cows were used throughout the entire study period to maintain consistency and reliability in the data collected.

### 2.3. Registration of Parameters

Throughout the experiment, the cows’ behavior parameters (rumination time min/d. (RT), body temperature (°C), reticulorumen pH, water consumption (L/day), cows’ activity (h/day)) were monitored using specialized SmaXtec boluses. The SmaXtec boluses (SmaXtec animal care GmbH, Graz, Austria) were administered orally at the beginning of the trial to cows within the first 30 days after calving. Boluses were applied in compliance with the guidelines provided by the manufacturer. First, buffer solutions with pH values of 4 and 7 were used to calibrate the pH, and after application, average data were recorded every ten minutes daily. The SmaXtec Messenger^®^ program gathered and displayed all the collected information. The rumen biosensor was administered orally by licensed veterinarians, and software (SmaXtec, Austria) was used to record rumen data. On the farm, a SmaXtec climate sensor (CS-5046, SmaXtec animal care GmbH, Belgiergasse 3, 8020 Graz, Austria) was used to register ambient humidity and temperature.

The hydrogen potential (pH), partial carbon dioxide pressure (pCO2), partial oxygen pressure (pO2), bicarbonate (HCO3), base excess in the extracellular fluid (BE (ecf)), oxygen saturation (sO2), sodium (Na+), potassium (K+), ionized calcium (Ca++), chlorides (Cl−), total carbon dioxide in the blood (TCO2), hematocrit (Hct), hemoglobin concentration (cHgb), base excess in the blood (BE), glucose (Glu), lactate (Lac), blood urea nitrogen (BUN), creatinine (Crea), and blood urea nitrogen to creatinine ratio (BUN/Crea) were measured.

Blood was collected from each cow using the jugular venipuncture procedure. To evaluate the acid-base balance, 1.6 mL of blood was collected using heparinized vacutainer tubes from Terumo Europe, located in Leuven, Belgium. Subsequently, the samples were identified and placed in an ice bath for a maximum duration of 30 min until the processing stage. We employed blood gas analyzers made by EPOC (Canada).

During the study period, the temperature–humidity index (THI) was recorded using a SmaXtec climate sensor (CS-5046, SmaXtec animal care GmbH, Graz, Austria) to measure ambient temperature and humidity. The THI was calculated using the following formula: THI = (0.8 × Tdb) + [(RH/100) × (Tdb − 14.4)] + 46.4 [23].

SmaXtec utilizes bolus technology to obtain data on the drinking behavior of individual cows by directly monitoring the reticulum. The bolus device monitors the internal body temperature and determines the amount of water consumed by utilizing AI-powered programs that analyze the temperature fluctuations following each drinking session. This enables the monitoring of each animal’s water intake to determine if it meets the desired levels, without requiring any further effort.

### 2.4. Creation of Groups

Based on the THI values, we categorized the data into four THI classes as follows: 1—THI 60–63 (4 June 2024–12 June 2024); 2—THI 65–69 (13 June 2024–18 June 2024); 3—THI 73–75 (19 June 2024–25 June 2024); and 4—THI 73–78 (26 June 2024–1 July 2024).

### 2.5. Duration of Parameter Registration

#### 2.5.1. Registration of Cow Behavior Parameters

On 1 June 2024, SmaXtec boluses were administered; the adaption phase lasted for 3 days. The measurements were completed on 1 July 2024, having begun on 4 June 2024. SmaXtec boluses were used to measure reticulorumen parameters such as temperature, pH, rumination index, and cow activity levels. Rumination activity levels over a 24 h period (24 h rolling sum) are displayed by the SmaXtec rumination meter and are totaled in minutes.

#### 2.5.2. Registration of Cows’ Blood Gas Parameters

In each THI class, blood gas parameters were registered once per group as follows: class 1 on 06.12, class 2 on 06.18, class 3 on 06.15, and class 4 on 07.01. The same parameters were recorded for the same cows.

### 2.6. Statistical Analysis

All the data were initially recorded using the SmaXtec Messenger^®^ program. Before analysis, the data were converted into a Microsoft Excel file. The Armonk, New York, USA-based IBM Corp. produced IBM SPSS Statistics for Windows, Version 25.0, (SPSS, 2017), which was used for the statistical analysis in this study. The data distribution’s normality was evaluated using the Shapiro–Wilk test [24] The standard error of the mean (SEM) and mean values of the results were presented, with the significance level set at 0.05 (*p* ≤ 0.05) to determine the existence of significant differences between the investigated parameters, such as rumination time, body temperature, reticulorumen pH, water consumption, cow activity, blood gas parameters, and temperature–humidity index. Additionally, the statistical relationships between the variables under study were examined using Pearson correlation analysis, with coefficients interpreted as follows: values between 0.1 and 0.3 indicated a low correlation, 0.3 to 0.5 indicated a moderate correlation, and values above 0.5 indicated a high correlation [25].

## 3. Results

### 3.1. Short-Term Effects of Heat Stress on Cow Behavior Registered by Innovative Technologies

#### 3.1.1. Short-Term Effects of Heat Stress on Cow Rumination Time

We found significant differences in rumination time between cows at lower risk of heat stress (THI 60–63), higher risk of heat stress (THI 73–75), and the highest risk of heat stress (THI 73–78). In the third group, rumination time was 16.17% lower (*p* < 0.001), and in the fourth group (THI 73–78), it was 70% lower (*p* < 0.001) compared to the first group. The average rumination time in the first group was 476.29 (±55.158) min/day, in the third group 399.57 (±103.74) min/day, and in the fourth group 139.36 (±19.44) min/day (Table 3).

#### 3.1.2. Short-Term Effects of Heat Stress on Cow Body Temperature

We found significant differences in body temperature between cows at lower risk of heat stress (THI 60–63) and those at the highest risk of heat stress (THI 73–78). The average body temperature in the first group of cows was 38.88 (±0.74) °C, while in the fourth group, it was 39.31 (±0.94) °C. The body temperature of the highest risk of heat stress group was 2% higher than in the cows with a lower risk of heat stress.

According to our results, we did not find any significant differences (*p* > 0.05) between the groups in reticulorumen pH, water consumption, and cow activity (Table 3).

### 3.2. Short-Term Effects of Heat Stress on Cow Blood Gas Parameters

#### 3.2.1. Short-Term Effects of Heat Stress on Cow on Partial Carbon Dioxide Pressure (pCO2)

We found significant differences in partial carbon dioxide pressure (pCO2) between the lower risk of heat stress group (THI 60–63) and the highest risk of heat stress group (THI 73–78). The average pCO2 in the first group was 34.91 (±4.74) mmHg, while in the fourth group it was 29.31 (±3.81) mmHg. pCO2 was 12.6% lower in the fourth group compared to the first group (Table 4).

#### 3.2.2. Short-Term Effects of Heat Stress on Cow on Partial Oxygen Pressure (pO2)

We found a 32% increase in partial oxygen pressure (pO2) in the group with a higher risk of heat stress. The average pO2 in the lower risk of heat stress group (THI 60–63) was 100.99 (±56.01) mmHg, while in the highest risk of heat stress group (THI 73–78) it was 149.86 (±38.16) mmHg (Table 4).

#### 3.2.3. Short-Term Effects of Heat Stress on Sodium (Na) Concentration in Cows

We found a 1.36% lower Na concentration in cows in the highest risk of heat stress group (THI 73–78). The average Na concentration in this group was 135.29 (±1.32) mmol/L, while in the lower risk of heat stress group (THI 60–63) it was 136.93 (±1.63) mmol/L (Table 4).

#### 3.2.4. Short-Term Effects of Heat Stress on Potassium (K) Concentration in Cows

We found a 6% lower potassium concentration in cows in the highest risk of heat stress group (THI 73–78) compared to the lower risk of heat stress group (THI 60–63). The average K concentration in the first group was 4.25 (±0.37) mmol/L, while in the fourth group it was 4.02 (±0.33) mmol/L (Table 4).

#### 3.2.5. Short-Term Effects of Heat Stress on Chloride (Cl) Concentration in Cows

We found an increase of 3% in Cl concentration in cows in the highest risk of heat stress group (THI 73–78) compared to the lower risk of heat stress group (THI 60–63). The average Cl concentration in the first group was 99.86 (±2.28) mmol/L, while in the fourth group it was 102.14 (±1.70) mmol/L (Table 4).

No significant differences (*p* > 0.05) were found between the cow groups in pH, cHCO3, BE (ecf), cSO2, Ca, TCO2, Hct, cHgb, Glu, Lac, BUN, urea, creatinine, and BUN/creatinine ratio (Table 4).

## 4. Discussion

According to the results of our earlier study, we recommend that dairy farmers utilize advanced technologies to continuously monitor and apply control heat stress management strategies in cows [26]. When the THI is above 78, it is important to carefully observe changes in activity time and chewing activities, and greater amounts of increases in these behaviors suggest greater levels of heat stress. Farmers should modify nutritional regimens in accordance with the ongoing monitoring of blood urea nitrogen levels. Utilizing cutting-edge technologies, this novel strategy has the potential to preserve the well-being and efficiency of dairy cows in different weather situations [27].

According to our present results, the short-term effects of heat stress on cow behavior can be registered by innovative technologies and blood gas parameters.

### 4.1. Short-Term Effects of Heat Stress on Cow Behavior Registered by Innovative Technologies

Compared to the THI 60–63 group, a reduction of 16.17% in rumination time was found in the THI 73–75 group and a decrease of 70% was found in the THI 73–78 group. The observed differences in rumination behavior across several HS risk categories emphasize the complex connection between environmental stresses and animal well-being [28] and heat stress has a detrimental impact on the respiratory rate in moderate climates, such as Germany in Central Europe [21]. Cows adjust their RT in response to certain heat stress thresholds, which are lower in moderate temperature zones compared to hotter places [29]. The greater reduction in RT is consistent with the findings published by Caja et al. (2016) [21] as well as the more recent study by Soriani et al. (2012) [30] during periods of summer heat stress in the Po valley (Italy). The impact of ambient temperature on the movement of the rumen and, consequently, on the functioning of the rumen that can increase rumination has been shown previously [31,32].

The categorization of heat stress in earlier investigations was based on the THI thresholds established by Zimbelman and Collier [33], which stated that a temperature–humidity index (THI) threshold of 68 is crucial in determining the impact on the milk production of cows in the southern United States. Nevertheless, Heinicke et al. (2018) [34] documented that HS can occur even at lower THI units in temperate climates. In Brandenburg, Germany, Heinicke et al. (2018) [34] established a heat load threshold of THI 67 based on the activity and milk yield of dairy cows. These findings align with the results of Bernabucci et al. (2010) [35], who found a decrease in milk production in Holstein Friesian cows throughout the first, second, and third lactations when the THI indices in Italy reached between 65 and 76. The authors emphasized that multiparous cows are more susceptible to heat stress compared to primiparous cows [35], which was confirmed by Gantner et al. (2011) [36]. A study conducted by Ammer et al. (2016) in Lower Saxony (France) reported that a THI of 65 was a key threshold that resulted in increase in reticular body temperature, using 28 Holstein Friesian cows [37].

In this study, the body temperature (BT) of the group at highest risk of heat stress (THI 73–78) exhibited a 2% increase compared with cows at a lower risk of heat stress (THI 60–63). BT is frequently utilized as a reliable indicator in practice to assess if a cow has achieved thermal equilibrium [8]. It is also employed to evaluate the impact of the thermal environment on the growth, lactation, and reproduction of dairy cows [8]. In this study, it was found that as the THI value increased from 45 to 72, the rectal temperature rose from 37.8 °C to 38.5°, which represented a 60% increase in THI and a 1.8% increase in BT. Nevertheless, while the THI value increased from 72 to 79, the BT increased from 38.5 °C to 39.35 °C, indicating a percentage rise of 9.7% and 2.2% for THI and RT, respectively. Put simply, when the THI exceeded 72, the percentage rise in BT was greater, suggesting a complex relationship between THI and BT. Thus, a THI value below 72 is regarded as the thermoneutral zone for dairy cows [38,39]. The BT of dairy cows was considerably greater during periods of HS compared to non HS periods (*p* < 0.01) [40]. The application of mechanical and electrical sensing technologies have led to a gradual improvement in automated monitoring equipment for measuring the BT and activity of cows [41]. In the future, these methods may play a crucial role in scientifically assessing HS in livestock by integrating THI, body temperature, and other indicators (such as activity), and these techniques can enhance the management of dairy cows, allowing for more precise and personalized approaches to predict heat stress efficiently [40]. Moreover, the increasing availability and accessibility of these and other emerging innovative technologies, such as wearable sensors and automated monitoring systems, are making it easier for farmers to adopt them. This widespread adoption is facilitated by ongoing advancements in technology, decreasing costs, and supportive policies from both national governments and the private sector, ensuring that even smaller-scale farmers can benefit from these tools.

### 4.2. Short-Term Effects of Heat Stress on Cow Blood Gas Parameters

There were notable disparities in partial carbon dioxide pressure (pCO2) between the THI groups in this study, with a reduced risk of heat stress (THI 60–63) and the group with the highest risk of heat stress (THI 73–78), and the fourth group (THI 73–78) had a 12.6% decrease in pCO2 compared to the first group.

The first hypocapnia was caused by the rise in respiratory rate (RR), as indicated by the strongest correlation between RR and THI during this period, which agreed with the studies by Garcia et al. [42] and West [8]. The increasing trend of RR has had a direct impact on pCO2 levels, as a reduction in pCO2 was found in cows experiencing heat stress during HS [43].

The group at greatest risk of HS (THI 73–78) exhibited a 32% rise in partial oxygen pressure (pO2), while the mean pO2 in the low heat stress risk group (THI 60–63) was 100.99 (±56.01) mmHg, but in the high HS risk group (THI 73–78) it was 149.86 (±38.16) mmHg. This result is consistent with the findings of Garcia et al. [42], who similarly observed found pO2 levels in cows under HS. According to Gupta et al. [1], the elevated RR would have enhanced the blood PO2 levels by increasing oxygen intake.

In the group of cows at the greatest risk of HS (THI 73–78), in this study had a lower sodium concentration of 1.36% compared with a reduced risk of HS (THI 60–63). The mean sodium concentration in this cohort was 135.29 (±1.32) mmol/L, whereas in the subgroup with a lower risk of HS (THI 60–63), that was 136.93 (±1.63) mmol/L. Greater ambient temperature leads to the excretion of Na+ ions by sweating and the study of Khuntia and Chaudhary (2002) [44] showed a notable rise in Na+ levels in heat-stressed cattle at a THI greater that 73, which further indicated that this increase is influenced by the animals’ water consumption, as this determines the extent of dehydration. Typically, in cases of HS, a significant amount of sodium (Na+) is lost by perspiration and salivation caused by panting [45].

There was a 6% decrease in potassium plasma concentration found in cows in the group at greatest risk of HS (THI 73–78) compared to the lower risk of HS group (THI 60–63). The mean potassium (K) plasma concentration in the lowest HS group (THI 60–63) was 4.25 (±0.37) mmol/L, whereas in the fourth group at greatest risk of HS (THI 73–78) it was 4.02 (±0.33) mmol/L. Burhans et al. (2022) [45] found that a significant quantity of potassium ions (K+) can be lost during periods of HS. Al-Qaisi et al. (2020) [46] found a notable reduction in potassium (K+) levels in ruminant animals, which was linked to the loss of this mineral through sweating during greater THI (THI 73–78). The farm’s effective nutritional management could have a significant impact in maintaining stable K+ levels during the early stage of lactation [47].

There was a 3% rise in Cl concentration in cows belonging to the group at greatest risk of HS (THI 73–78) compared to the group at lower risk of heat stress (THI 60–63). The mean Cl content in the first group was 99.86 (±2.28) mmol/L, but in the fourth group (THI 73–78) it was 102.14 (±1.70) mmol/L. When there is a requirement to restore normal pH levels by the compensatory removal of HCO3−, the retention of Cl− can result in an elevation in Cl− levels [48]. Do Nguyen et al. (2022) [48] showed elevated plasma Cl concentrations in cows experiencing HS and it was noted that the loss of HCO3−, which is responsible for regulating respiratory alkalosis, is accompanied by the retention of Cl− due to the Cl shift phenomenon occurring in renal tubules. In this situation, the maintenance of normal levels of both Cl− and HCO3− indicates that the kidneys may not be fully functional in terms of regulating the acid-base electrolyte balance. However, Joo et al. (2021) [49] showed a decline in chloride (Cl−) concentrations as a result of HS in Holstein Friesian and Jersey cows. The preservation of plasma Cl− and HCO3− concentrations within the normal range suggests that the body’s acid-base equilibrium was well controlled throughout the research period. The transient elevation in blood pH caused by hypocapnia did not seem to induce substantial disruptions in the animal’s physiological equilibrium, hence suggesting the effectiveness of the nutritional supplements implemented on the farm in this study [47].

To mitigate the adverse effects of heat stress, dairy farm managers should consider implementing strategies that are informed by the study’s findings. For instance, the continuous monitoring of rumination time and body temperature using advanced technologies, such as SmaXtec boluses, can serve as early indicators of heat stress. Farmers can then take timely actions, such as increasing ventilation, providing shade, and ensuring adequate water supply, to prevent the escalation of stress in cows [8]. Moreover, adjusting feeding schedules to cooler parts of the day [50] and incorporating feed additives that promote heat tolerance [51] could further enhance the resilience of dairy cattle to heat stress. The integration of these practices into regular farm management could improve animal welfare and sustain milk production even under challenging environmental conditions [52].

While this study provides valuable data on the short-term effects of heat stress, future research should explore its long-term impacts and evaluate the effectiveness of various mitigation strategies. Additionally, developing cost-effective and user-friendly monitoring technologies will be crucial for wider adoption by small and medium-sized farms. Collaborations between researchers, technology providers, and policymakers will be essential to ensure these tools are accessible and effectively integrated into dairy farm management.

## 5. Conclusions

The results of this study showed that HS lowered rumination time by up to 70% in Lithuanian Holstein dairy cows within the greatest THI classification group (73–78) and increased BT by 2%, with a 12.6% lower partial carbon dioxide pressure (pCO2) and a 32% increase in partial oxygen pressure (pO2), also decreasing sodium by 1.36% and potassium by 6%, while increasing chloride by 3%.

Veterinarians, farmers, and other stakeholders should employ advanced monitoring technologies to detect early signs of HS, such as changes in rumination time (lowered by up to 70%) and BT (increased by 2%). Additionally, blood gas parameters including partial carbon dioxide pressure (pCO2), partial oxygen pressure (pO2), sodium (Na), potassium (K), and chloride (Cl) can be monitored to manage the physiological impacts of HS.

An additional recommendation to national governments and the private sector is to prioritize making these innovative technologies accessible and affordable, particularly for the dairy industry in hot climates. This can be achieved through subsidies, grants, and financial incentives, as well as partnerships with technology providers to reduce costs. Such measures will help ensure that dairy farmers and other stakeholders can effectively manage heat stress, improve animal welfare, and maintain productivity even in challenging environmental conditions.

## Figures and Tables

**Table 1 animals-14-02390-t001:** Diet composition for a 620 kg Holstein cow.

Diet Component	Percentage (%)
Corn silage	25
Alfalfa grass hay	5
Grass silage	20
Sugar beet pulp silage	15
Grain concentrate mash	30
Mineral mix	5

**Table 2 animals-14-02390-t002:** Nutrient composition of the ration.

Nutrient	Composition (%) or Mcal/kg
Dry matter (DM)	48.8
Neutral detergent fiber	28.2
Acid detergent fiber	19.8
Non-fiber carbs	38.7
Crude protein	15.8
Net lactation energy	1.6 Mcal/kg

**Table 3 animals-14-02390-t003:** Descriptive statistics of cow behavior parameters, registered by innovative technologies.

Parameter	THI Class	N	Mean	Std. Deviation	95% Confidence Interval for Mean		Minimum	Maximum	*p*
					Lower Bound	Upper Bound			
Rumination time (min/d.)	1 ^a^	14	476.29	55.15	444.44	548	380	548	0.67
	2 ^b^	14	455.93	75.16	412.53	536	334	536	0.56
	3 ^c^	14	399.57 ^a^	103.74	339.67	528	133	528	0.02
	4 ^d^	14	139.36 ^a^	19.44	128.13	172	107	172	0.001
	Total	56	342.77	191.46	291.50	548	37	548	
Body temperature (°C)	1 ^a^	14	38.88	0.74	38.45	40	38	40	0.07
	2 ^b^	14	38.77	0.86	38.27	40	37	40	0.14
	3 ^c^	14	38.85	1.19	37.16	40	36	40	0.64
	4 ^d^	14	39.31 ^b^	0.94	38.77	40	37	40	0.02
	Total	56	30.38	14.22	26.57	40	5	40	
Reticulorumen pH	1 ^a^	14	6.14	0.33	5.95	7	6	7	0.56
	2 ^b^	14	6.14	0.28	5.97	7	6	7	0.28
	3 ^c^	14	6.13	0.33	5.93	7	6	7	0.65
	4 ^d^	14	6.02	0.42	5.77	7	5	7	0.51
	Total	56	39.44	58.97	23.65	172	6	172	
Water consumption (L/day)	1 ^a^	13	130.62	17.91	119.79	164	103	164	0.87
	2 ^b^	13	131.77	12.93	123.95	160	113	160	0.53
	3 ^c^	14	133.14	22.32	120.25	172	73	172	0.41
	4 ^d^	14	173.20	0.00	73.20	73	73	73	0.07
	Total	54	100.31	56.03	85.02	172	3	172	
Cow activity (h/day)	1 ^a^	13	8.77	5.21	5.62	19	2	19	0.43
	2 ^b^	13	9.43	5.25	6.25	21	2	21	0.07
	3 ^c^	14	9.61	4.38	7.08	18	1	18	0.25
	4 ^d^	14	10.12	4.62	7.45	19	3	19	0.52
	Total	54	25.85	28.57	18.05	73	1	73	

THI Class—1—THI 60–63 (4 June 2024–12 June 2024); 2—THI 65–69 (13 June 2024–18 June 2024); 3—THI 73–75 (19 June 2024–25 June 2024); and 4—THI 73–78 (26 June 2024–1 July 2024). N—number of cows; *p*—coefficient of significance. The letters a, b, c and d indicate statistically significant differences between HS groups (*p* ≤ 0.05).

**Table 4 animals-14-02390-t004:** Descriptive statistics of cow blood gas parameters.

Parameter	THI Class	N	Mean	Std. Deviation	95% Confidence Interval for Mean		Minimum	Maximum	*p*
pCO2 (mmHg)	1 ^a^	14	34.91	4.74	32.17	43	28	43	0.45
	2 ^b^	14	33.94	5.91	30.53	47	22	47	0.32
	3 ^c^	13	34.27	4.61	31.48	47	30	47	0.57
	4 ^d^	14	29.31 ^a^	3.81	27.10	38	25	38	0.01
	Total	55	33.09	5.21	31.68	47	22	47	
pO2 (mmHg)	1 ^a^	14	100.99	56.01	68.64	190	38	190	0.17
	2 ^b^	14	135.24	63.75	98.43	214	43	214	0.67
	3 ^c^	13	129.42	54.30	96.60	200	54	200	0.76
	4 ^d^	14	149.86 ^a^	38.16	127.83	194	81	194	0.01
	Total	55	128.87	55.37	113.90	214	38	214	
cHCO3	1 ^a^	14	26.99	1.90	25.89	30	23	30	0.54
	2 ^b^	14	25.44	2.73	23.86	31	19	31	0.09
	3 ^c^	13	25.82	1.87	24.69	29	22	29	0.67
	4 ^d^	14	25.03	1.76	24.01	28	22	28	0.54
	Total	55	25.82	2.18	25.23	31	19	31	
BE (ecf)	1 ^a^	14	3.80	2.36	2.44	8	−2	8	0.34
	2 ^b^	14	2.05	2.42	0.65	7	−4	7	0.79
	3 ^c^	13	2.42	2.14	1.12	6	−2	6	0.09
	4 ^d^	14	2.52	2.00	1.36	6	−1	6	0.34
	Total	55	2.70	2.28	2.09	8	−4	8	
cSO2	1 ^a^	14	92.44	9.46	86.97	100	76	100	0.09
	2 ^b^	14	96.20	5.73	92.89	100	83	100	0.45
	3 ^c^	13	97.53	3.24	95.57	100	91	100	0.76
	4 ^d^	14	99.28	0.77	98.83	100	97	100	0.43
	Total	55	96.34	6.20	94.66	100	76	100	
Na (mmol/L)	1 ^a^	14	136.93	1.63	135.98	140	134	140	0.32
	2 ^b^	14	136.29	1.63	135.34	139	133	139	0.76
	3 ^c^	13	135.85	1.14	135.16	138	134	138	0.52
	4 ^d^	14	135.29 ^a^	1.32	134.52	138	133	138	0.02
	Total	55	136.09	1.54	135.67	140	133	140	
K (mmol/L)	1 ^a^	14	4.25	0.37	4.04	5	4	5	0.56
	2 ^b^	14	4.24	0.19	4.13	5	4	5	0.87
	3 ^c^	13	4.41	0.38	4.18	5	4	5	0.54
	4 ^d^	14	4.02 ^a^	0.33	4.10	5	4	5	0.03
	Total	55	4.30	0.32	4.21	5	4	5	
Ca (mmol/L)	1 ^a^	14	1.12	0.04	1.09	1	1	1	0.67
	2 ^b^	14	1.13	0.03	1.11	1	1	1	0.82
	3 ^c^	13	1.17	0.05	1.14	1	1	1	0.41
	4 ^d^	14	1.15	0.03	1.13	1	1	1	0.62
	Total	55	1.14	0.04	1.13	1	1	1	
Cl (mmol/L)	1 ^a^	14	99.86	2.28	98.54	102	94	102	0.54
	2 ^b^	14	101.43	1.69	100.45	103	96	103	0.62
	3 ^c^	13	101.15	2.19	99.83	105	98	105	0.87
	4 ^d^	14	102.14 ^a^	1.70	101.16	107	100	107	0.03
	Total	55	101.15	2.10	100.58	107	94	107	
TCO2 (mmHg)	1 ^a^	14	26.09	1.80	25.05	29	23	29	0.56
	2 ^b^	14	25.16	2.94	23.46	31	18	31	0.67
	3 ^c^	13	25.55	1.97	24.36	29	22	29	0.85
	4 ^d^	14	24.61	1.83	23.55	27	21	27	0.32
	Total	55	25.35	2.20	24.76	31	18	31	
Hct (% fraction)	1 ^a^	14	24.50	1.22	23.79	26	22	26	0.56
	2 ^b^	14	24.21	1.76	23.20	27	21	27	0.76
	3 ^c^	13	23.46	1.94	22.29	27	21	27	0.41
	4 ^d^	14	23.64	1.86	22.57	26	20	26	0.73
	Total	55	23.96	1.72	23.50	27	20	27	
cHgb (g/dL)	1 ^a^	14	8.34	0.48	8.06	9	7	9	0.07
	2 ^b^	14	8.24	0.58	7.90	9	7	9	0.09
	3 ^c^	13	7.93	0.65	7.54	9	7	9	0.13
	4 ^d^	14	8.00	0.61	7.65	79	7	9	0.67
	Total	55	8.13	0.59	7.97	9	7	9	
Glu (mmol/L)	1 ^a^	14	2.54	0.44	2.28	4	2	4	0.74
	2 ^b^	14	3.01	0.45	2.75	4	2	4	0.57
	3 ^c^	13	2.78	0.30	2.59	3	2	3	0.56
	4 ^d^	14	3.36	0.32	3.17	4	3	4	0.32
	Total	55	2.92	0.48	2.79	4	2	4	
Lac (mmol/L)	1 ^a^	14	2.66	1.38	1.86	5	1	5	0.87
	2 ^b^	14	1.36	0.94	0.81	3	0	3	0.08
	3 ^c^	13	2.16	0.94	1.59	5	1	5	0.09
	4 ^d^	14	0.88	0.79	0.42	3	0	3	0.76
	Total	55	1.76	1.23	1.42	5	0	5	
BUN (g/L)	1 ^a^	14	12.07	2.61	10.56	16	6	16	0.56
	2 ^b^	14	11.86	2.62	10.34	17	7	17	0.32
	3 ^c^	13	12.38	2.21	11.04	17	8	17	0.09
	4 ^d^	14	10.07	2.16	8.82	15	6	15	0.18
	Total	55	11.58	2.52	10.90	17	6	17	
Urea (g/L)	1 ^a^	14	4.32	0.93	3.78	6	2	6	0.09
	2 ^b^	14	4.24	0.91	3.71	6	3	6	0.76
	3 ^c^	13	4.46	0.74	4.01	6	3	6	0.67
	4 ^d^	14	3.54	0.82	3.06	6	2	6	0.76
	Total	55	4.13	0.91	3.89	6	2	6	
Crea (mmol/L)	1 ^a^	14	74.07	11.52	67.42	98	56	98	0.67
	2 ^b^	14	70.79	9.13	65.51	87	54	87	0.52
	3 ^c^	13	78.46	11.42	71.56	98	61	98	0.72
	4 ^d^	14	72.21	8.80	67.13	85	59	85	0.41
	Total	55	73.80	10.39	70.99	98	54	98	
Bun/Crea	1 ^a^	14	14.65	3.33	12.73	21	10	21	0.65
	2 ^b^	14	15.11	3.87	12.88	24	9	24	0.55
	3 ^c^	13	14.38	2.98	12.57	18	9	18	0.42
	4 ^d^	14	12.23	2.74	10.64	18	8	18	0.83
	Total	55	14.09	3.36	13.18	24	8	24	
pH	1 ^a^	14	7.50	0.06	7.46	8	7	8	0.09
	2 ^b^	14	7.49	0.04	7.46	8	7	8	0.45
	3 ^c^	13	7.49	0.04	7.46	8	7	8	0.32
	4 ^d^	14	7.54	0.04	7.51	8	7	8	0.74
	Total	55	7.50	0.05	7.49	8	7	8	

THI Class—1—THI 60–63 (4 June 2024–12 June 2024); 2—THI 65–69 (13 June 2024–18 June 2024); 3—THI 73–75 (19 June 2024–25 June 2024); and 4—THI 73–78 (26 June 2024–1 July 2024).). pH—hydrogen potential; pCO2—partial carbon dioxide pressure; pO2—partial oxygen pressure; HCO3—bicarbonate; BE (ecf)—base excess in the extracellular fluid; sO2—oxygen saturation; Na+—sodium; K+—potassium; Ca++—ionized calcium; Cl−—chlorides; TCO2—total carbon dioxide in the blood; Hct—hematocrit; cHgb—hemoglobin concentration; BE—base excess in the blood; Glu—glucose; Lac—lactate; BUN—blood urea nitrogen; Crea—creatinine; BUN/Crea—blood urea nitrogen and creatinine ratio. The letters a, b, c and d indicate statistically significant differences between HS groups (*p* ≤ 0.05). N—number of cows; *p*—coefficient of significance.

## Data Availability

The data provided in this study can be found in the publication.

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
