# Peer review of "Short-Term Effects of Heat Stress on Cow Behavior, Registered by Innovative Technologies and Blood Gas Parameters"

_animals, 2024, doi:10.3390/ani14162390_

Round 1
Reviewer 1 Report
Comments and Suggestions for Authors
The paper titled "Short-Term Effects of Heat Stress on Cow Behavior, Registered by Innovative Technologies and Blood Gas Parameters" addresses the significant issue of heat stress on dairy cows and its impact on their behavior and physiological parameters. Here is a summary of the key points and some suggestions for improvement:
The paper examines how heat stress affects cow behavior and blood gas parameters, using advanced monitoring technologies. This topic is highly relevant given the increasing global temperatures and their potential impact on livestock.
Comments and Suggestions:
I suggest rewriting the simple summary. According to the author's guidelines, this section should summarize and contextualize your paper within the existing literature in your field. It should be written without technical language or nonstandard acronyms, with the goal of conveying the meaning and importance of this research to non-experts.
I recommend rewriting the abstract and including more results and the significance of the obtained data.
Introduction:
The introduction provides a solid background on the importance of heat stress in dairy cows. However, it would benefit from a more detailed review of previous studies on similar topics to better situate the current research within the broader context. I suggest citing: 10.1016/j.vas.2024.100363 regarding the heat stress conditions and 10.3168/jdsc.2020-0074 and 10.1080/1828051X.2021.1884005 for rumen pH boluses detection and rumen tag collars.
Methodology:
The methodology section is comprehensive but could be improved by including more details on the statistical methods used for data analysis. For example, specify the types of statistical tests used to determine the significance of the findings.
Clarify the selection criteria for the 56 cows used in the study out of the 1070 clinically evaluated. This will help readers understand the representativeness of the sample.
The authors should consider including references to support the statistical methods used in the analysis. Some key areas to address include: 10.29261/pakvetj/2020.067 for shapiro wilk test, the main statistical model is missing, and 10.3389/fvets.2024.1332207 for pearson correlation.
Results:
While the results are clearly presented, the section would benefit from additional graphical representations of the data. Graphs or charts depicting the changes in behavior and blood gas parameters across different THI classes would make the findings more accessible and visually appealing.
Discussion:
The discussion effectively interprets the findings but could delve deeper into the implications of these results for dairy farm management practices. Providing specific recommendations for mitigating heat stress based on the study's findings would add practical value.
Compare and contrast the study's results with those of other similar studies to highlight unique contributions or confirm existing knowledge.
Conclusion:
The conclusion succinctly summarizes the key findings but could be strengthened by emphasizing the broader significance of the study for the dairy industry and potential future research directions.
References: Ensure all references are up-to-date and relevant. It would also be useful to include more recent studies that align with the innovative technologies discussed.
Clarity and Precision: Some sections, particularly those describing technical details, could be written more clearly to ensure they are accessible to a broader audience, including those not specialized in veterinary sciences.
Author Response
Dear Reviewer,
The authors are grateful for your helpful comments and suggestions. We have revised the entire manuscript according to your feedback. All corrections have been highlighted using track changes and marked in yellow.
Best regards,
Prof. Dr. Ramunas Antanaitis
Comments and Suggestions: I suggest rewriting the simple summary. According to the author's guidelines, this section should summarize and contextualize your paper within the existing literature in your field. It should be written without technical language or nonstandard acronyms, with the goal of conveying the meaning and importance of this research to non-experts.
Response: We corrected simple summary part to – “This study examines how heat stress affects the behavior and health of dairy cows using advanced monitoring technologies. We found that higher levels of heat stress significantly reduce the time cows spend ruminating and increase their body temperature. Furthermore, key blood gas parameters are altered, indicating changes in the cows' metabolic state. These findings highlight the importance of using innovative technologies to monitor and manage heat stress in dairy cows to ensure their health and productivity”
Comments and Suggestions I recommend rewriting the abstract and including more results and the significance of the obtained data.
Response: we corrected whole abstract section.
Comments and Suggestions: Introduction:
The introduction provides a solid background on the importance of heat stress in dairy cows. However, it would benefit from a more detailed review of previous studies on similar topics to better situate the current research within the broader context. I suggest citing: 10.1016/j.vas.2024.100363 regarding the heat stress conditions and 10.3168/jdsc.2020-0074 and 10.1080/1828051X.2021.1884005 for rumen pH boluses detection and rumen tag collars.
Response: We corrected to – „Regarding physiological factors, there were stronger positive connections seen between respiratory rate and rectal temperature and both barn temperature and THI [16]. „
Comments and Suggestions: Methodology:
The methodology section is comprehensive but could be improved by including more details on the statistical methods used for data analysis. For example, specify the types of statistical tests used to determine the significance of the findings.
Response: we corrected – “The Armonk, New York, USA-based IBM Corp. produced IBM SPSS Statistics for Windows, Version 25.0, (SPSS, 2017), which was used for the statistical analysis in this study. The data distribution's normality was evaluated using the Shapiro-Wilk test [23] The standard error of the mean (SEM) and mean values of the results were presented, with the significance level set at 0.05 (p £ 0.05) to determine the levels of significance. Additionally, the statistical relationships between the variables under study were examined using Pearson correlation analysis [24]. “
Comments and Suggestions: Clarify the selection criteria for the 56 cows used in the study out of the 1070 clinically evaluated. This will help readers understand the representativeness of the sample.
Response: we corrected to – “A total of 56 of the 1070 clinically evaluated Lithuanian Holstein cows—that is, those in their second or subsequent lactation and within the first 30 days following calving—were chosen for the study. These cows were selected because they are in a critical period for lactation performance and are more likely to experience heat stress, which could impact their milk production and overall health. Additionally, cows in their second or subsequent lactations provide a consistent basis for evaluating the effects of heat stress due to their established lactation history. These cows weighed an average of 620 kg ± 45 kg, and they produced 12,700 kg of energy-corrected milk per lactation (with 4.2% fat and 3.6% protein). The same cows were used throughout the entire study period to maintain consistency and reliability in the data collected.”
Comments and Suggestions:The authors should consider including references to support the statistical methods used in the analysis. Some key areas to address include: 10.29261/pakvetj/2020.067 for shapiro wilk test, the main statistical model is missing, and 10.3389/fvets.2024.1332207 for pearson correlation.
Response: We corrected this section to – „The Armonk, New York, USA-based IBM Corp. produced IBM SPSS Statistics for Windows, Version 25.0, in 2017, which was used for the statistical analysis in this study. The data distribution's normality was evaluated using the Shapiro-Wilk test [22] The standard error of the mean (SEM) and mean values of the results were presented, with the significance level set at 0.05 (p < 0.05) to determine the levels of significance. Additionally, the statistical relationships between the variables under study were examined using Pearson correlation analysis [23]. „
Comments and Suggestions: Results:
While the results are clearly presented, the section would benefit from additional graphical representations of the data. Graphs or charts depicting the changes in behavior and blood gas parameters across different THI classes would make the findings more accessible and visually appealing.
Response: We deleted these figures because they repeated the results presented in the table. For clarity, we split the table into two separate tables. Additionally, we structured the results section into different subsections.
Comments and Suggestions: Discussion:
The discussion effectively interprets the findings but could delve deeper into the implications of these results for dairy farm management practices. Providing specific recommendations for mitigating heat stress based on the study's findings would add practical value.
Compare and contrast the study's results with those of other similar studies to highlight unique contributions or confirm existing knowledge.
Response: We corrected discusion section and added – „To mitigate the adverse effects of heat stress, dairy farm managers should consider implementing strategies that are informed by the study’s findings. For instance, continuous monitoring of rumination time and body temperature using advanced technologies, such as SmaXtec boluses, can serve as early indicators of heat stress. Farmers can then take timely actions, such as increasing ventilation, providing shade, and ensuring adequate water supply, to prevent the escalation of stress in cows [25]. Moreover, adjusting feeding schedules to cooler parts of the day [26] and incorporating feed additives that promote heat tolerance [27] could further enhance the resilience of dairy cattle to heat stress. The integration of these practices into regular farm management could improve animal welfare and sustain milk production even under challenging environmental conditions [28].
While this study provides valuable data on the short-term effects of heat stress, future research should explore long-term impacts and evaluate the effectiveness of various mitigation strategies. Additionally, developing cost-effective and user-friendly monitoring technologies will be crucial for wider adoption by small and medium-sized farms. Collaborations between researchers, technology providers, and policymakers will be essential to ensure these tools are accessible and effectively integrated into dairy farm management.“
Comments and Suggestions: Conclusion:
The conclusion succinctly summarizes the key findings but could be strengthened by emphasizing the broader significance of the study for the dairy industry and potential future research directions.
Response: we corrected conclusion section – “Based on our results, heat stress significantly reduces rumination time by up to 70% in cows within the highest THI class (73-78) and increases body temperature by 2% and a 12.6% lower in partial carbon dioxide pressure (pCO2), a 32% increase in partial oxygen pressure (pO2), and decrease sodium by 1.36% and potassium by 6%, while increasing chloride by 3% in the highest THI class.
Veterinarians, farmers and other stakeholders should employ advanced monitoring technologies to detect early signs of heat stress, such as changes in rumination time and body temperature. Additionally, track specific blood gas parameters including partial carbon dioxide pressure (pCO2), partial oxygen pressure (pO2), sodium (Na), potassium (K), and chloride (Cl) can be to monitored to manage the physiological impacts of heat stress.
An additional recommendation to national governments and the private sector is to prioritize making these innovative technologies accessible and affordable, particularly for the dairy industry in hot climates. This can be achieved through subsidies, grants, and financial incentives, as well as partnerships with technology providers to reduce costs. Such measures will help ensure that dairy farmers and other stakeholders can effectively manage heat stress, improve animal welfare, and maintain productivity even in challenging environmental conditions.“
Comments and Suggestions: References: Ensure all references are up-to-date and relevant. It would also be useful to include more recent studies that align with the innovative technologies discussed.
Response: We made corrections according to your recommendations and those of the other reviewers.
Comments and Suggestions: Clarity and Precision: Some sections, particularly those describing technical details, could be written more clearly to ensure they are accessible to a broader audience, including those not specialized in veterinary sciences.
Response: We made corrections according to your recommendations and those of the other reviewers. Thanks for comments and sugestions.
Reviewer 2 Report
Comments and Suggestions for Authors
I have indicated my comments and suggestions for the authors in the attached review report.

In general, the quality of the English Language is satisfactory and allows readers to appreciate what the authors are communicating. A few errors however have been pointed out as indicated in the attached review report.
Author Response
Dear Reviewer,
The authors are grateful for your helpful comments and suggestions. We have revised the entire manuscript according to your feedback. All corrections have been highlighted using track changes and marked in yellow. All answer are provided in your table as comments.
Best regards,
Prof dr, Ramunas Antanaitis

Reviewer 3 Report
Comments and Suggestions for Authors
This paper needs a good deal of work on the analysis and presentation. See attached file.

Needs considerable improvement before publication can be considered.
Author Response
Dear Reviewer,
The authors are grateful for your helpful comments and suggestions. We have revised the entire manuscript according to your feedback. All corrections have been highlighted using track changes and marked in yellow. All answers are provided in pdf as comments.
Best regards,
Prof dr, Ramunas Antanaitis

Round 2
Reviewer 1 Report
Comments and Suggestions for Authors
the paper improved a lot, no more suggestions from my side
Author Response
Dear Reviewer,
Thank you for your feedback. I appreciate your time and effort in reviewing the paper, and I'm glad to hear that it has improved significantly. Your comments have been invaluable in enhancing the quality of the work.
Best regards,
Prof. Ramunas Antanaitis
Reviewer 3 Report
Comments and Suggestions for Authors
The manuscript still needs revision of the English.
It may help to use grammar checking software in the prepartion of furure submisions.

The manuscript still needs revision of the English.
It may help these authors to use grammar checking software in the prepartion of furure submisions.
Author Response
Dear Reviewer,
Thank you for your valuable feedback and thorough review. We have carefully addressed all the comments and made the necessary corrections to the manuscript. A detailed response to each comment has been provided in the attached PDF file. We greatly appreciate your insights, which have significantly improved the quality of the paper. Thank you for your time and consideration.
Best regards,
Prof. Ramunas Antanaitis
